# Contrastive learning enhanced retrieval-augmented few-shot framework for multi-label patent classification

Wenlong Zheng[1], Xin Li[2]*, Guoqing Cui[2,3], Shikun Chen[1]

1 Ningbo University of Finance and Economics, School of Finance and Information, Ningbo, Zhejiang, China, 2 The First Topographic Surveying Brigade of Ministry of Natural Resource of P.R.C., Xi'an, Shaanxi, China, 3 Northwest Land and Resources Research Center, Shaanxi Normal University, Xi'an, Shaanxi, China

* LiXin_DXYD@hotmail.com

## Abstract

The rapid expansion of patent databases poses increasing challenges for multi-label patent classification, particularly for inventions spanning multiple technological domains. Conventional approaches are hindered by high annotation costs and limited scalability, while often neglecting the semantic structure of patent documents. Here, we present a retrieval-enhanced few-shot learning framework that combines patent-specific contrastive pre-training with semantic retrieval to enable scalable multi-label classification. Drone technologies are selected as the evaluation domain due to their multidisciplinary characteristics encompassing mechanical, electronic, and software aspects. The proposed method learns domain-adapted embeddings that capture multi-label co-occurrence patterns and leverages retrieval-augmented few-shot learning with structured reasoning to reduce reliance on extensive annotations. Experiments on a curated dataset of 15,000 annotated drone patents across ten categories demonstrate that the framework achieves Macro-F1 and Micro-F1 scores of 0.847 and 0.892, corresponding to improvements of 30% and 23% over few-shot baselines. Furthermore, contrastive pre-training yields notable benefits for underrepresented categories, with performance improvements reaching 16% over transformer-based approaches. These results indicate that the proposed approach offers an effective and resource-efficient solution for multi-label patent classification, with potential to improve the scalability and accessibility of intellectual property analysis.

## Introduction

The global intellectual property landscape operates through advanced classification systems that organize technological innovations across diverse fields. International patent offices, including the United States Patent and Trademark Office, the European Patent Office, and the World Intellectual Property Organization, collectively manage large repositories containing tens of millions of patent documents [1].

reproduction in any medium, provided the original author and source are credited.

**Data availability statement:** All data and code are available at https://github.com/redcican/Contrastive-learning-latent-multi-label-classification.

**Funding:** Supported by the Fundamental Research Funds for the Central Universities, CHD; Project Number: 30010234450201.

**Competing interests:** The authors have declared that no competing interests exist.

These repositories rely on hierarchical classification frameworks such as the International Patent Classification and Cooperative Patent Classification systems, which structure technological knowledge into detailed categories that enable efficient prior art searches and strategic innovation analysis [2].

Contemporary patent filing rates have reached unprecedented levels. As a result, conventional classification methods face increasing difficulty, since the volume of new applications exceeds the capacity of manual review by several orders of magnitude [3]. This exponential growth intersects with increasing technological complexity, as modern patents frequently span multiple domains simultaneously. Rather than fitting neatly into single categories, today's patents often encompass interdisciplinary concepts that challenge conventional classification boundaries [4].

The technical challenges inherent in patent classification extend beyond mere scale considerations. Patents exhibit an inherently multi-label nature, with individual documents simultaneously belonging to multiple technological categories [5]. Consider autonomous vehicle patents: they encompass sensor technology, artificial intelligence algorithms, mechanical engineering principles, and telecommunications components within unified systems. This complexity is compounded by the specialized vocabulary and technical precision required in patent language, where subtle terminological differences can indicate entirely different technological approaches [6].

The annotation bottleneck represents a particularly acute challenge in this domain. Unlike general text classification where crowd-sourcing can provide adequate labels, patent classification requires deep technical expertise across multiple disciplines. This makes large-scale annotation efforts economically prohibitive [7]. Traditional supervised approaches demand thousands of labeled examples per category, while the specialized knowledge required for accurate patent labeling severely limits the pool of qualified annotators [8]. A fundamental tension thus exists between the need for adequate training data and the practical constraints of obtaining expert-level annotations.

Few-shot learning paradigms offer promising solutions for addressing these annotation constraints. Their application to specialized technical domains, however, presents distinct challenges due to complex semantic relationships and domain-specific terminology [9]. Rather than relying on general-purpose language representations, technical documents demand approaches that capture domain-specific semantic nuances. Recent advances in contrastive learning have shown promise for addressing representation challenges in specialized domains by learning embeddings that capture semantic relationships between similar and dissimilar samples [10].

Existing contrastive frameworks typically focus on single-label scenarios and fail to account for the complex co-occurrence patterns that characterize multi-label technical documents [11]. Retrieval-augmented approaches have simultaneously gained attention as powerful tools for enhancing few-shot performance by leveraging semantic similarity to identify relevant demonstration examples [12]. The combination of these techniques presents untapped potential for technical document classification, particularly in scenarios where label relationships exhibit hierarchical or co-occurrence structures.

Contrastive pre-training provides a natural complement to retrieval-based approaches by learning representations that explicitly encode semantic relationships between documents [13]. In the patent domain, contrastive objectives can be designed to respect multi-label co-occurrence patterns, ensuring that patents sharing technological categories are embedded in similar representation spaces while maintaining separation between distinct domains [14]. Such an approach enables retrieval mechanisms to identify demonstration examples that are not only semantically similar but also relevant for multi-label prediction tasks.

Unmanned aerial vehicle (UAV) technologies exemplify the multi-label classification challenges inherent in modern patent analysis. UAV innovations typically integrate mechanical components such as propulsion systems and structural elements, electronic systems including sensors and control circuits, software algorithms for navigation and autonomous control, and communication technologies for data transmission and remote control [15]. This technological convergence creates patent documents that naturally span multiple classification categories, providing an ideal testbed for multi-label few-shot learning approaches.

The framework presented in this study integrates contrastive pre-training with retrieval-enhanced few-shot learning to address multi-label patent classification under minimal annotation constraints. Our approach employs patent-specific contrastive objectives that capture multi-label co-occurrence patterns while leveraging semantic retrieval to identify demonstration examples that guide few-shot classification decisions. Through chain-of-thought reasoning, the framework methodically evaluates each potential label by considering retrieved examples and learned representations, enabling structured multi-label predictions without extensive training data.

Our work makes several contributions to the intersection of few-shot learning and technical document classification. We introduce a contrastive pre-training strategy specifically designed for multi-label patent classification that respects technological co-occurrence patterns while learning discriminative representations. We develop a retrieval-enhanced few-shot framework that leverages semantic similarity to identify informative demonstration examples for multi-label prediction tasks. Through comprehensive evaluation on UAV patent data encompassing ten technological categories, we demonstrate that our approach achieves meaningful improvements over conventional few-shot baselines while requiring minimal labeled training data.

## Background and related work

### Patent classification with domain-specific language models

Patent classification constitutes a cornerstone of intellectual property management, yet the exponential growth of patent filings has rendered traditional classification approaches increasingly inadequate. The complexity of patent documents – characterized by specialized technical vocabulary, legal terminology, and interdisciplinary content – poses formidable challenges for automated classification systems [16]. Modern patents frequently span multiple technological domains simultaneously, necessitating multi-label classification frameworks rather than single-label approaches [17].

The landscape of patent classification has witnessed significant evolution through the adoption of deep learning architectures. Transformer-based models, particularly those leveraging pre-trained language representations, have achieved notable success in capturing the nuanced semantics of patent text [18]. PatentNet employs fine-tuned variants of BERT, XLNet, and RoBERTa for multi-label patent classification, establishing new benchmarks on the USPTO-2M dataset [5]. These advances notwithstanding, such approaches demand extensive labeled training data–a requirement that proves particularly onerous given the specialized expertise necessary for accurate patent annotation [19].

Domain-specific language models have emerged as a promising direction for addressing these challenges. PatentGPT and PatentSBERTa represent significant advances in patent-specific pre-training, with PatentGPT trained on over 240 billion tokens of patent-related text demonstrating superior performance on intellectual property benchmarks [18,20,21]. Continual pre-training strategies offer cost-effective domain adaptation by leveraging existing general-purpose models as initialization points [22,23]. The construction of domain-specific training corpora requires careful curation, incorporating

patent specifications, office actions, prior art citations, and technical literature [24–26]. Vocabulary adaptation to accommodate technical terms, chemical formulas, and specialized notation further improves both compression rates and semantic representation quality [27,28].

The multi-label nature of patent classification introduces additional complexities beyond traditional document categorization. Patents describing autonomous systems may simultaneously encompass mechanical engineering, artificial intelligence, telecommunications, and control systems–each warranting distinct classification labels [29]. Hierarchical approaches have been proposed to capture structured relationships in patent classification systems, with graph convolutional networks effectively modeling label dependencies in the International Patent Classification hierarchy [30]. UAV patents exemplify these multi-disciplinary challenges, spanning mechanical, electronic, software, and communication technologies [31,32]. This technological convergence makes UAV patents an ideal testbed for evaluating multi-label classification methods designed to handle complex, interdisciplinary innovations.

## Contrastive learning for document classification

Contrastive learning has revolutionized representation learning across various domains by explicitly optimizing for discriminative embeddings that cluster similar samples while separating dissimilar ones [33]. In the context of document classification, this paradigm shift has yielded notable improvements, particularly in scenarios with limited labeled data. The fundamental principle–learning representations through the lens of similarity and dissimilarity–aligns naturally with the objectives of classification tasks [34].

Recent advances have extended contrastive frameworks to accommodate the complexities of multi-label scenarios. Traditional contrastive objectives, designed primarily for single-label settings, fail to capture the nuanced relationships present when documents can simultaneously belong to multiple categories [35]. CAROL (Class-Aware Contrastive Loss) addresses this limitation by incorporating class separation objectives specifically tailored for imbalanced multi-label text classification [36]. Similarly, label-aware contrastive approaches have demonstrated superior performance by explicitly modeling inter-label relationships during the representation learning phase [37].

The application of contrastive learning to specialized technical domains presents unique opportunities and challenges. Technical documents exhibit domain-specific semantic structures that differ markedly from general text corpora [38]. SimCSE (Simple contrastive learning of sentence embeddings), while successful in general sentence embedding tasks, requires careful adaptation when applied to patent text, where subtle terminological distinctions carry significant legal and technical implications [39]. Recent work has shown that domain-adapted contrastive objectives, which incorporate technical term relationships and hierarchical concept structures, outperform generic contrastive frameworks [40].

The synergy between contrastive pre-training and downstream classification tasks has proven particularly effective. CICA (Content-Injected Contrastive Alignment) demonstrates that incorporating document-specific content modules during contrastive training enhances zero-shot classification capabilities [35]. This approach suggests that contrastive objectives can be designed not merely to learn general representations, but to actively prepare models for specific downstream tasks. For patent classification, this implies the possibility of designing contrastive objectives that respect the multi-label co-occurrence patterns inherent in technological innovation [14].

Decoupled supervised contrastive learning represents another promising direction, separating the representation learning phase from the classification objective [41]. This decoupling allows for more stable training dynamics and improved convergence, particularly beneficial when dealing with the high-dimensional, sparse label spaces characteristic of patent classification [42]. The approach has shown remarkable success in domains requiring fine-grained discrimination between semantically similar categories – a common requirement in patent analysis [43].

## Retrieval-augmented few-shot learning

The integration of retrieval mechanisms with few-shot learning paradigms has emerged as a powerful approach for addressing data scarcity challenges. Atlas, a landmark contribution in this area, demonstrates that retrieval-augmented

language models can achieve competitive performance on knowledge-intensive tasks with minimal training examples [44]. By leveraging external knowledge sources during both training and inference, these models effectively augment their limited parametric knowledge with non-parametric retrieval [45–47].

The application of retrieval-augmented approaches to classification tasks requires careful consideration of demonstration selection strategies. Rather than relying on random sampling from limited training sets, retrieval-based methods identify semantically relevant examples that provide maximal information for classification decisions [48]. This targeted selection proves particularly valuable in technical domains where examples must capture domain-specific nuances [49]. Recent work has shown that task-specific retrieval metrics, learned jointly with classification objectives, outperform generic similarity measures [50,51]. In-context learning approaches have demonstrated effectiveness for text classification with many labels, where demonstration selection proves crucial for handling large label spaces [52]. Few-shot learning has also achieved notable advances in computer vision tasks, particularly multimodal approaches for 3D point cloud segmentation [53,54]. These cross-domain developments may inform future methodological innovations in patent classification.

Meta-learning frameworks have been successfully combined with retrieval mechanisms to enhance few-shot performance. By meta-training on diverse tasks while incorporating retrieval-based demonstration selection, models develop the ability to rapidly adapt to new domains with minimal examples [55]. This approach proves especially relevant for patent classification, where new technological categories emerge continuously and obtaining extensive labeled data for each category remains impractical [55].

The quality and relevance of retrieved demonstrations influence few-shot classification performance. RePrompt introduces visual prompt learning enhanced by retrieval mechanisms, demonstrating that carefully curated retrieval databases can bridge domain gaps in few-shot scenarios [56]. For patent classification, this suggests that building domain-specific retrieval corpora–containing technically relevant demonstrations – could significantly enhance classification accuracy even with limited labeled examples [57].

Retrieval-augmented approaches also address the challenge of handling rare or emerging categories. In patent classification, where technological innovations continuously create new subcategories, the ability to leverage similar historical patents as demonstrations becomes invaluable [58]. QZero demonstrates that query reformulation through retrieval can improve zero-shot classification by enriching input representations with relevant contextual information [56].

**Chain-of-thought reasoning for multi-label prediction**

Chain-of-thought (CoT) prompting has fundamentally transformed how language models approach complex reasoning tasks. By decomposing multi-step problems into intermediate reasoning steps, CoT enables models to tackle challenges that would otherwise exceed their capabilities [59]. This structured reasoning approach proves particularly relevant for multi-label classification, where decisions about individual labels may depend on complex inter-label relationships [60].

The application of CoT to classification tasks extends beyond simple prompting strategies. Recent work demonstrates that explicitly modeling the reasoning process for each label decision improves both accuracy and interpretability [61]. In multi-label scenarios, this translates to systematic evaluation of each potential label, considering both positive and negative evidence before making classification decisions [11]. The transparency afforded by chain-of-thought reasoning also facilitates error analysis and model improvement [62].

Zero-shot chain-of-thought approaches have shown effectiveness when combined with retrieval mechanisms. By generating reasoning chains based on retrieved examples, models can perform complex classification without task-specific training [63]. This capability proves especially valuable in patent classification, where the reasoning behind label assignments often involves technical comparisons and legal considerations that benefit from explicit articulation [8].

The emergence of chain-of-thought reasoning as an inherent capability of sufficiently large models suggests interesting implications for domain-specific applications [64]. While general-purpose models demonstrate this capability on

arithmetic and commonsense reasoning tasks, adapting CoT for specialized domains like patent classification requires careful consideration of domain-specific reasoning patterns [65]. The technical nature of patent documents demands reasoning chains that incorporate technological relationships, innovation boundaries, and classification hierarchy constraints [29].

## Materials and methods

### Dataset

Our experimental foundation rests upon a large patent corpus sourced from the National Intellectual Property Administration's IP Search and Consultation Center China. This repository contains one million patmporal scope captures the evolution of contemporary technologies, while the volume provides sufficient diversity for robust few-shot learning evaluation.

From this extensive collection, we extracted and curated approximately 100,000 patents related to unmanned aerial vehicle technologies, with 15,000 receiving expert annotations across ten technological categories. This curated UAV patent dataset represents a contribution of this work and is available at https://github.com/redcican/Contrastive-learning-latent-multi-label-classification. The extraction methodology employed a multi-stage filtering approach combining domain-specific keyword identification with controlled vocabulary validation. Our filtering strategy incorporated 47 carefully selected terms organized across five categorical domains: general UAV terminology, configuration-specific descriptors, functional classifications, application-specific identifiers, and technical component designators.

The validation process consisted of three sequential stages designed to ensure extraction precision while minimizing noise. Initial keyword-based filtering identified candidate patents using Boolean search operators to capture documents containing multiple relevant concepts. Cross-validation against Cooperative Patent Classification codes, particularly B64C, B64D, and B64U categories, provided systematic verification of technological relevance. Manual expert review of a stratified sample comprising 2,000 patents assessed both precision and recall, achieving 94.2% and 89.6% respectively.

Quality control measures addressed potential domain leakage and terminological ambiguity through several mechanisms. Automated removal eliminated patents with conflicting or unclear abstracts, while expert review resolved boundary cases where multiple technological domains intersected. Negative filtering terms excluded non-technical applications such as recreational models and simulation software. This systematic approach resulted in a curated dataset with verified technological relevance and minimal noise contamination.

Patent abstracts constitute the primary textual input for classification models due to their comprehensive yet concise summaries of technological developments. The preprocessing pipeline transforms these abstracts through sequential cleaning, translation, and normalization steps. Since many patents originated in Chinese, neural machine translation via Google Translate API generated English versions for linguistic consistency. Text cleaning removed patent-specific boilerplate phrases, special characters, and residual markup. Tokenization employed whitespace delimiters followed by lemmatization to preserve semantic meaning while reducing vocabulary complexity.

The classification framework targets ten distinct technological categories that capture the multidisciplinary nature of UAV innovations. These categories emerged through systematic thematic analysis combining inductive corpus exploration with expert domain knowledge. Initial topic modeling using Latent Dirichlet Allocation on 10,000 patent abstracts revealed 15 preliminary themes. Expert consolidation by aerospace engineering specialists merged semantically related topics while eliminating overly granular categories. Hierarchical clustering analysis validated topic coherence and ensured adequate categorical separation. The multi-label co-occurrence patterns inherent in this schema provide rich training signals for contrastive pre-training objectives, enabling the model to learn representations that capture technological relationships and domain-specific semantic structures.

Table 1 presents representative examples demonstrating the technological diversity within our dataset. Each entry illustrates the multidisciplinary nature of UAV innovations, from bionic flight mechanisms to modular deployment systems.

**Table 1**. Representative patent examples from the UAV technology dataset.

| UAV Technology Patent Examples | | |
| --- | --- | --- |
| **Technology Focus** | **Patent Title** | **Abstract Excerpt** |
| Bionic Systems | Multi-modal flapping-wing aircraft for agricultural applications | This invention discloses a bio-inspired UAV system incorporating multi-modal quadrotor deployment platforms with flapping-wing propulsion mechanisms designed for precision agricultural operations... |
| Power Optimization | Solar-integrated endurance platform | A novel UAV design featuring flexible photovoltaic arrays integrated into wing structures for extended flight duration. The system incorporates advanced power management algorithms and hybrid energy storage... |
| Modular Design | Rapid-deployment emergency response drone | This patent describes a modular UAV architecture enabling quick assembly and reconfiguration for emergency scenarios. The system features interchangeable payload modules and collapsible structural elements... |
| VTOL Systems | Hybrid fixed-wing transition aircraft | An innovative aircraft combining vertical take-off capabilities with fixed-wing cruise efficiency. The design incorporates variable-geometry wing structures and distributed propulsion systems... |
| Surveillance Applications | Multi-sensor reconnaissance platform | This invention presents an advanced surveillance UAV equipped with multi-spectral imaging sensors, radar systems, and real-time data processing capabilities for intelligence gathering operations... |

The target classification schema includes the following technological categories: VTOL and hybrid flight systems focusing on vertical take-off capabilities and transition mechanisms; bionic and flapping wing designs mimicking biological flight patterns; modular and deployable architectures emphasizing rapid assembly and compact storage; endurance and power optimization systems incorporating advanced energy solutions; structural integrity and material developments addressing durability and performance; surveillance, inspection, and mapping applications covering data collection and monitoring; logistics and cargo transport capabilities; multi-environment operations including amphibious and submersible functionalities; flight control and stability systems featuring advanced algorithms and mechanisms; and specialized applications addressing niche requirements such as agricultural or emergency operations.

A binary relevance approach transforms the multi-label classification problem into independent binary decisions for each technological category. This formulation maintains compatibility with contrastive learning objectives while facilitating efficient computation during few-shot learning scenarios. The dataset structure supports few-shot learning through carefully balanced class representations and diverse technological examples that enable effective demonstration retrieval across all categories. Ground truth labels were established through structured annotation by three independent domain experts, each possessing over five years of specialized experience in aerospace engineering and patent analysis.

Inter-annotator agreement analysis using Fleiss' Kappa achieved a score of 0.82, indicating substantial consensus among experts and validating annotation reliability. When disagreements occurred in approximately 12% of cases, a senior patent analyst with over ten years of experience facilitated consensus discussions to establish definitive classifications. This structured annotation process ensures the validity and reliability of our experimental evaluation framework. The curated dataset serves as both training data for contrastive pre-training and a retrieval corpus for identifying semantically similar demonstration examples during few-shot classification tasks.

Table 2 summarizes key characteristics of our curated dataset, including class distributions and textual properties.

## Contrastive pre-training for multi-label patent representations

Given a patent corpus $\mathcal{D} = \{(x_i, y_i)\}_{i=1}^{N}$ where $x_i$ represents patent abstract text and $y_i \in \{0, 1\}^K$ denotes the multi-label assignment across $K$ technological categories, the objective is to learn an encoder $f_\theta : \mathcal{X} \to \mathbb{R}^d$ that maps patent text to

**Table 2.** Dataset characteristics and technological category distributions.

| UAV Patent Dataset Statistics | | |
|---|---|---|
| **Characteristic** | **Seed Dataset** | **Complete Dataset** |
| Total annotated patents | 5,000 | 15,000 |
| Target categories | 10 | 10 |
| Average labels per patent | 2.2 | 2.3 |
| Average abstract length | 182 words | 185 words |
| Vocabulary size | 18,500 | 24,700 |
| **Technology Category Distribution** | | |
| VTOL & Hybrid Flight | 2,280 | 6,850 |
| Surveillance & Mapping | 1,970 | 5,900 |
| Flight Control & Stability | 1,630 | 4,900 |
| Modular & Deployable | 1,030 | 3,100 |
| Endurance & Power Systems | 930 | 2,800 |
| Structural & Materials | 830 | 2,500 |
| Logistics & Cargo | 720 | 2,150 |
| Bionic & Flapping Wing | 650 | 1,950 |
| Specialized Applications | 520 | 1,550 |
| Multi-Environment | 320 | 950 |

Notes: Category counts represent patent frequencies in seed and final datasets. Totals exceed dataset size due to multi-label assignments where patents belong to multiple technological categories.

$d$-dimensional embeddings. The encoder consists of a RoBERTa-large backbone [66] followed by projection layers that transform contextualized representations into contrastive embedding space.

The multi-label contrastive framework extends standard contrastive objectives by incorporating label co-occurrence relationships. For a given anchor patent $x_i$ with labels $y_i$, we define positive examples as patents sharing at least one technological category: $\mathcal{P}_i = \{j : y_i \cap y_j \neq \varnothing, j \neq i\}$. Rather than treating all other patents as negatives, we introduce a similarity-weighted approach that accounts for partial label overlap.

The multi-label contrastive loss combines instance-level and label-aware objectives:

$$\mathcal{L}_{contrast} = \mathcal{L}_{instance} + \lambda \mathcal{L}_{label} \tag{1}$$

where $\lambda$ controls the balance between instance and label-level learning. The instance-level loss employs InfoNCE with multi-label positive sampling:

$$\mathcal{L}_{instance} = -\frac{1}{N} \sum_{i=1}^{N} \log \frac{\sum_{j \in \mathcal{P}_i} \exp(\text{sim}(z_i, z_j)/\tau)}{\sum_{k=1}^{N} \exp(\text{sim}(z_i, z_k)/\tau)} \tag{2}$$

where $z_i = f_\theta(x_i)$ represents the embedding for patent $i$, $\text{sim}(z_i, z_j) = z_i^T z_j/(\|z_i\|\|z_j\|)$ denotes cosine similarity, and $\tau$ is the temperature parameter.

The label-aware component explicitly models technological co-occurrence patterns through a weighted similarity measure. We compute label similarity between patents as:

$$s_{label}(y_i, y_j) = \frac{|y_i \cap y_j|}{|y_i \cup y_j|} \cdot \exp(-\alpha|y_i \triangle y_j|) \tag{3}$$

where $|y_i \triangle y_j|$ represents symmetric difference and $\alpha$ penalizes large label disparities. The label-aware loss becomes:

$$\mathcal{L}_{label} = -\frac{1}{N}\sum_{i=1}^{N}\frac{1}{\sum_{j\neq i}s_{label}(y_i,y_j)}\sum_{j\neq i}s_{label}(y_i,y_j)\log\frac{\exp(\text{sim}(z_i,z_j)/\tau)}{\sum_{k\neq i}\exp(\text{sim}(z_i,z_k)/\tau)} \qquad (4)$$

Patent-specific adaptations address domain characteristics including technical terminology density and hierarchical category relationships. We incorporate domain-adaptive temperature scaling:

$$\tau_i = \tau_0 \cdot (1 + \beta \cdot \text{complexity}(x_i)) \qquad (5)$$

where complexity is measured by technical term frequency. For hierarchical categories, we introduce category-aware negative sampling that reduces the probability of selecting patents from semantically related categories as hard negatives.

The training procedure employs momentum-based updates to maintain stable positive pairs across mini-batches. For each training step, we construct balanced mini-batches ensuring adequate representation from all technological categories. The momentum encoder $f_{\theta'}$ with slowly updated parameters:

$$\theta' = m\theta' + (1-m)\theta \qquad (6)$$

provides consistent positive examples across iterations where $m$ is the momentum coefficient. Training involves three phases: (1) initialization using RoBERTa-large weights, (2) contrastive pre-training on the patent corpus with frozen RoBERTa parameters, and (3) joint fine-tuning of both RoBERTa and projection layers. This progressive training strategy balances domain adaptation with preservation of general linguistic knowledge. Fig 1 illustrates the contrastive pre-training architecture and multi-label similarity computation process.

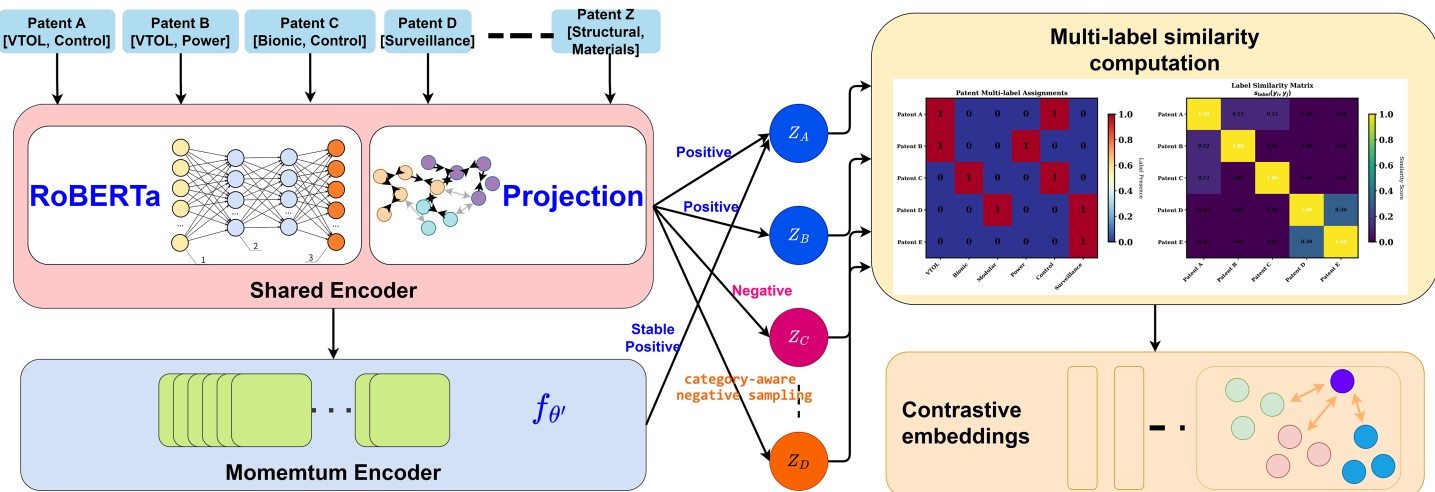

**Fig 1**. **Contrastive pre-training architecture for multi-label patent representations.** The system processes patent abstracts through a shared encoder to generate embeddings that are optimized using multi-label contrastive objectives. The similarity computation considers both instance-level relationships and label co-occurrence patterns, enabling the model to learn representations that capture technological relationships and domain-specific semantic structures.

## Retrieval-augmented demonstration selection

The retrieval-augmented demonstration selection mechanism leverages the contrastive embeddings learned during pre-training to identify semantically relevant patent examples that guide few-shot classification decisions. Given a query patent $x_q$ requiring multi-label classification, the retrieval system searches through a corpus of labeled patents $\mathcal{C} = \{(x_j, y_j)\}_{j=1}^{M}$ to identify demonstrations that maximize both semantic similarity and label informativeness.

The retrieval process operates in the embedding space created by the pre-trained encoder $f_\theta$. For a query patent $x_q$, we compute its embedding $z_q = f_\theta(x_q)$ and retrieve the $k$ most similar patents based on a multi-faceted similarity metric:

$$\text{score}(x_q, x_j) = \alpha_1 \cdot \text{sim}_{sem}(z_q, z_j) + \alpha_2 \cdot \text{sim}_{tech}(x_q, x_j) + \alpha_3 \cdot \text{div}(x_j | \mathcal{R}) \tag{7}$$

where $\text{sim}_{sem}(z_q, z_j)$ represents semantic similarity in the contrastive embedding space, $\text{sim}_{tech}(x_q, x_j)$ captures technical domain alignment through specialized features, and $\text{div}(x_j | \mathcal{R})$ promotes diversity among retrieved examples to avoid redundancy.

The semantic similarity component leverages the contrastive embeddings directly:

$$\text{sim}_{sem}(z_q, z_j) = \frac{z_q^T z_j}{\|z_q\| \|z_j\|} \cdot \exp\left(-\gamma \cdot d_{edit}(x_q, x_j)\right) \tag{8}$$

where $d_{edit}(x_q, x_j)$ represents an edit distance penalty that down-weights near-duplicate patents, preventing the retrieval of trivially similar examples that provide limited discriminative information.

Technical domain alignment incorporates patent-specific features including International Patent Classification (IPC) codes, technical term overlap, and citation relationships:

$$\text{sim}_{tech}(x_q, x_j) = w_1 \cdot \text{IPC}_{overlap}(x_q, x_j) + w_2 \cdot \text{term}_{match}(x_q, x_j) + w_3 \cdot \text{cite}_{rel}(x_q, x_j) \tag{9}$$

where weights $w_1, w_2, w_3$ are learned during validation to optimize retrieval quality for the specific patent domain.

To ensure retrieved demonstrations provide comprehensive coverage of the label space, we introduce a diversity-promoting mechanism that penalizes redundant retrievals:

$$\text{div}(x_j | \mathcal{R}) = 1 - \max_{x_r \in \mathcal{R}} \left( \text{sim}_{sem}(z_j, z_r) \cdot \frac{|y_j \cap y_r|}{|y_j \cup y_r|} \right) \tag{10}$$

where $\mathcal{R}$ represents the set of already retrieved examples and the Jaccard similarity $\frac{|y_j \cap y_r|}{|y_j \cup y_r|}$ measures label overlap, preventing retrieval of examples with redundant label combinations.

The retrieval strategy adapts dynamically based on query characteristics. For patents with high technical complexity (measured by specialized term density), we increase the weight of technical similarity:

$$\alpha_2^{adaptive} = \alpha_2 \cdot \left( 1 + \delta \cdot \frac{\text{tech\_terms}(x_q)}{\text{avg\_tech\_terms}} \right) \tag{11}$$

This adaptation ensures that highly technical patents receive demonstrations from similar technological domains, improving the relevance of retrieved examples.

To handle the multi-label nature of patent classification, we implement a label-aware retrieval strategy that considers the co-occurrence patterns learned during contrastive pre-training. The retrieval process prioritizes examples that exhibit

similar label complexity:

$$\text{label\_match}(x_q, x_j) = \exp\left(-\eta \cdot \left| |y_q^{pred}| - |y_j| \right|\right) \cdot \prod_{l \in y_j} p(l|z_q) \tag{12}$$

where $|y_q^{pred}|$ represents the predicted number of labels for the query (estimated from embedding characteristics), and $p(l|z_q)$ denotes the probability of label $l$ given the query embedding, computed using a lightweight classifier head.

The final retrieval set $\mathcal{D}_k = \{(x_i, y_i)\}_{i=1}^{k}$ consists of the top-$k$ patents according to the combined scoring function. These demonstrations are ordered by relevance and formatted as input-output pairs that illustrate the multi-label classification task. The ordering preserves semantic coherence while ensuring label diversity:

$$\mathcal{D}_k^{ordered} = \arg \text{sort}_{\mathcal{D}_k} \left[ \lambda_1 \cdot \text{score}(x_q, x_i) - \lambda_2 \cdot \sum_{j<i} \text{sim}_{sem}(z_i, z_j) \right] \tag{13}$$

This ordering strategy balances individual relevance with collective diversity, creating a demonstration set that provides comprehensive guidance for multi-label prediction. Fig 2 illustrates the retrieval-augmented demonstration selection process and its integration with the few-shot prediction module.

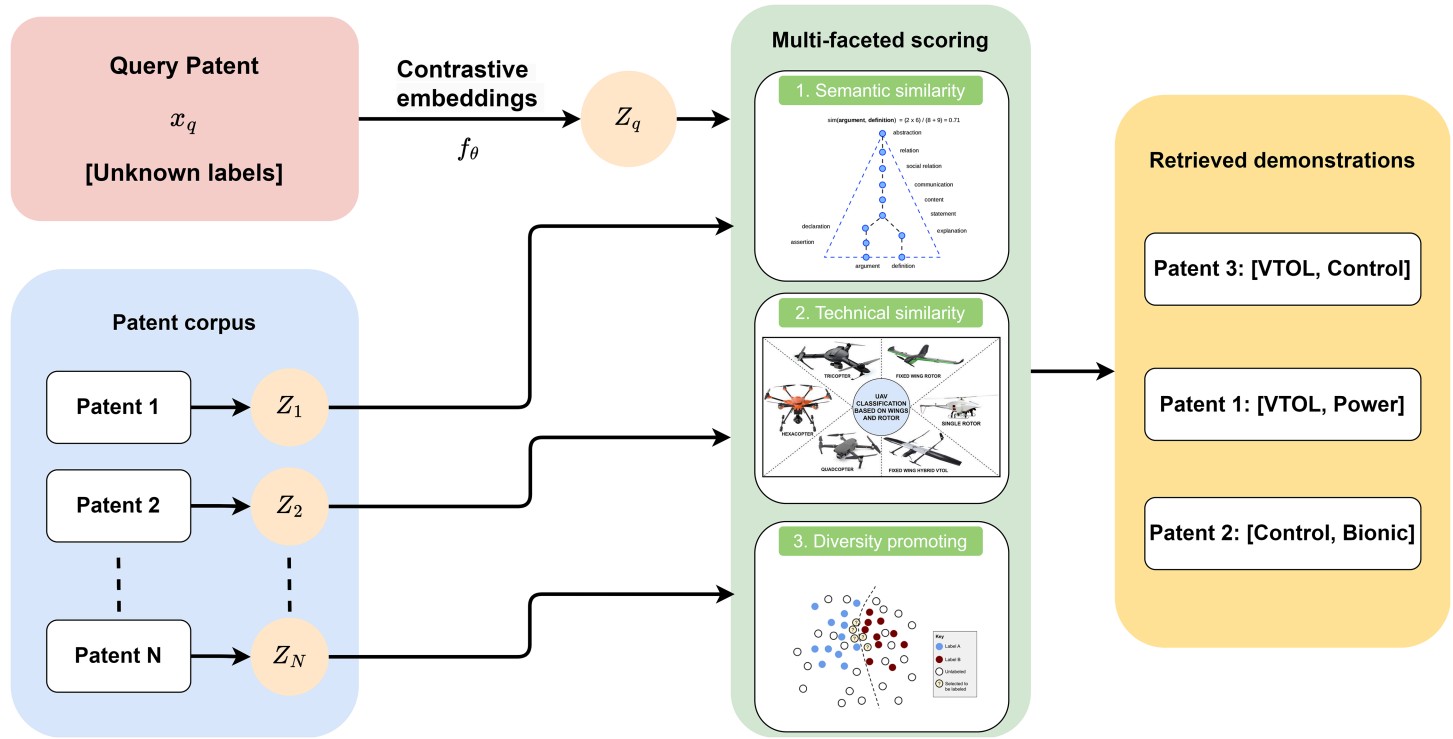

**Fig 2**. **Retrieval-augmented demonstration selection.** The system leverages contrastive embeddings to identify relevant patent demonstrations through multi-faceted similarity scoring. The retrieval process considers semantic similarity, technical domain alignment, and diversity constraints to select informative examples that guide multi-label classification decisions. Retrieved demonstrations are ordered to balance relevance and diversity, providing comprehensive coverage of the label space.

**Few-shot multi-label prediction**

The few-shot prediction module integrates the contrastive embeddings and retrieved demonstrations to classify patents across multiple categories using minimal labeled examples. Given a query patent $x_q$ and its retrieved demonstration set $\mathcal{D}_k^{ordered} = \{(x_i, y_i)\}_{i=1}^k$, the system constructs a structured prompt that enables systematic multi-label prediction through in-context learning.

The prompt construction follows a task-specific template that explicitly frames the multi-label classification problem:

$$\mathcal{P}(x_q, \mathcal{D}_k) = \text{Instruct} \oplus \bigoplus_{i=1}^k \text{Demo}(x_i, y_i) \oplus \text{Query}(x_q) \tag{14}$$

where $\oplus$ denotes concatenation, Instruct provides task instructions emphasizing multi-label nature, $\text{Demo}(x_i, y_i)$ formats each demonstration as an input-output pair, and $\text{Query}(x_q)$ presents the target patent for classification.

Each demonstration is formatted to highlight the multi-label assignment pattern:

$$\text{Demo}(x_i, y_i) = \text{``Patent''} : x_i \rightarrow \text{``Categories''} : \{l_j : y_i^j = 1, j \in [1, K]\} \tag{15}$$

This explicit formatting helps the model recognize that multiple categories can be simultaneously assigned, distinguishing the task from single-label classification scenarios.

To leverage the learned representations, we introduce an embedding-guided attention mechanism that modulates the influence of each demonstration based on its relevance to the query:

$$w_i = \text{softmax}\left(\frac{\text{sim}(z_q, z_i)}{\tau_{attn}} + b_i\right) \tag{16}$$

where $z_q$ and $z_i$ are contrastive embeddings, $\tau_{attn}$ is an attention temperature parameter, and $b_i$ represents a bias term that accounts for demonstration position and label informativeness:

$$b_i = \alpha_{pos} \cdot \exp(-\beta_{pos} \cdot i) + \alpha_{label} \cdot |y_i \cap y_q^{pred}| \tag{17}$$

The position-dependent term gives higher weight to earlier demonstrations (which are more relevant according to our retrieval ordering), while the label overlap term prioritizes demonstrations sharing predicted labels with the query.

For multi-label prediction, we employ a decomposed inference strategy that evaluates each category independently while considering inter-label dependencies:

$$p(y_q^j = 1 | x_q, \mathcal{D}_k) = \sigma\left(f_{LM}(x_q, \mathcal{D}_k, l_j) + \sum_{m \neq j} \omega_{jm} \cdot \mathbb{1}[y_q^m = 1]\right) \tag{18}$$

where $f_{LM}$ represents the language model's output for category $l_j$, $\sigma$ is the sigmoid function, and $\omega_{jm}$ captures learned dependencies between categories $j$ and $m$.

The language model scoring function $f_{LM}$ combines multiple sources of evidence:

$$f_{LM}(x_q, \mathcal{D}_k, l_j) = \text{logit}_{LM}(l_j | \mathcal{P}) + \lambda_1 \cdot \text{sim}_{cat}(x_q, l_j) + \lambda_2 \cdot \text{prior}(l_j | z_q) \tag{19}$$

where $\text{logit}_{LM}(l_j|\mathcal{P})$ is the language model's raw score for category $l_j$ given the prompt, $\text{sim}_{cat}(x_q, l_j)$ measures category-specific similarity using learned category prototypes, and $\text{prior}(l_j|z_q)$ provides a prior probability based on the contrastive embedding.

To handle the varying complexity of multi-label assignments, we implement an adaptive threshold mechanism:

$$\tau_j^{thresh} = \tau_0 + \Delta\tau \cdot \left(\text{freq}(l_j) - \mu_{freq}\right) + \gamma \cdot \text{uncertainty}(x_q, l_j) \tag{20}$$

where $\text{freq}(l_j)$ is the category frequency in the retrieval corpus, $\mu_{freq}$ is the mean frequency, and $\text{uncertainty}(x_q, l_j)$ quantifies prediction uncertainty based on demonstration consistency.

The uncertainty measure evaluates the consistency of label assignments across retrieved demonstrations:

$$\text{uncertainty}(x_q, l_j) = -p_j \log(p_j) - (1 - p_j)\log(1 - p_j) \text{ where } p_j = \sum_{i=1}^{k} w_i \cdot y_i^j \tag{21}$$

High uncertainty (inconsistent demonstrations) leads to more conservative thresholds, reducing false positive predictions. For categories with sparse representation in the demonstration set, we employ a prototype-based fallback mechanism:

$$\text{proto}_j = \frac{1}{|\mathcal{S}_j|} \sum_{x \in \mathcal{S}_j} f_\theta(x) \tag{22}$$

where $\mathcal{S}_j$ represents the support set of patents containing category $l_j$. When demonstration coverage for category $j$ is insufficient ($\sum_{i=1}^{k} y_i^j < \epsilon$), the prediction incorporates prototype similarity:

$$p_{fallback}(y_q^j = 1) = \sigma\left(\frac{z_q^T \text{proto}_j}{\|z_q\| \|\text{proto}_j\| \tau_{proto}}\right) \tag{23}$$

The final prediction combines demonstration-based and prototype-based predictions:

$$\hat{y}_q^j = \mathbb{1}\left[\alpha_{demo} \cdot p(y_q^j = 1|x_q, \mathcal{D}_k) + (1 - \alpha_{demo}) \cdot p_{fallback}(y_q^j = 1) > \tau_j^{thresh}\right] \tag{24}$$

where $\alpha_{demo} = \min(1, \sum_{i=1}^{k} y_i^j / \epsilon)$ weights the contribution based on demonstration coverage.

This approach ensures robust multi-label predictions even with limited demonstrations, leveraging both the semantic understanding from contrastive pre-training and the pattern recognition from retrieved examples. The adaptive mechanisms handle the inherent challenges of multi-label classification, including label imbalance, sparse categories, and varying label co-occurrence patterns.

## Chain-of-thought multi-label reasoning

The chain-of-thought (CoT) reasoning mechanism enhances the few-shot prediction process by introducing structured reasoning paths for multi-label decisions. Rather than directly predicting labels from demonstrations, the system employs GPT-4o to generate explicit reasoning chains that evaluate each technological category through systematic analysis of patent characteristics.

We implement GPT-4o integration through OpenAI's API with carefully tuned parameters for patent classification. The API calls utilize the GPT-4o model with temperature set to 0.3 for consistent reasoning while maintaining creativity,

max_tokens of 2048 to accommodate detailed reasoning chains, top_p of 0.9 for controlled diversity in technical terminology, and frequency_penalty of 0.2 to reduce repetitive patterns in multi-label evaluations. The system prompt explicitly defines the patent classification task and multi-label nature, while maintaining conversation context across category evaluations to preserve inter-label dependencies. Response format is set to JSON mode to ensure structured output parsing for downstream processing.

Given the prompt $\mathcal{P}(x_q, \mathcal{D}_k)$ constructed in the previous step, we augment it with reasoning instructions that decompose the multi-label classification into sequential evaluation steps:

$$\mathcal{P}_{CoT} = \mathcal{P}(x_q, \mathcal{D}_k) \oplus \text{Reason}_{template} \tag{25}$$

where $\text{Reason}_{template}$ instructs the model to: (1) identify key technological features in the query patent, (2) compare these features against each demonstration's label assignments, (3) evaluate evidence for each category independently, and (4) synthesize final multi-label predictions.

The reasoning process for each category $l_j$ follows a structured template:

$$\text{CoT}_j = \text{Extract}(x_q, l_j) \rightarrow \text{Compare}(\mathcal{D}_k, l_j) \rightarrow \text{Evaluate}(evidence_j) \rightarrow \text{Decide}(y_q^j) \tag{26}$$

This decomposition enables the model to articulate why specific labels apply, improving both accuracy and interpretability.

To handle inter-label dependencies systematically, the reasoning incorporates conditional evaluation:

$$p_{CoT}(y_q^j = 1 | x_q, \mathcal{D}_k, \mathcal{Y}_{<j}) = \text{GPT-4o}\left(\mathcal{P}_{CoT}, l_j, \{y_q^i : i < j\}\right) \tag{27}$$

where $\mathcal{Y}_{<j}$ represents previously assigned labels, allowing the model to consider technological relationships. For instance, if "VTOL" is assigned, the reasoning for "Flight Control" explicitly considers this context.

The CoT mechanism integrates with the embedding-guided attention by using attention weights $w_i$ to emphasize reasoning about highly relevant demonstrations:

$$\text{Reason}_{weighted} = \sum_{i=1}^{k} w_i \cdot \text{CoT}(x_q, x_i, y_i) \tag{28}$$

This ensures that reasoning focuses on the most informative examples identified through retrieval. For sparse categories where demonstrations are limited, the reasoning explicitly incorporates prototype comparisons:

$$\text{CoT}_{sparse} = \text{"Since few examples exist for } l_j\text{, comparing query to category prototype..."} \tag{29}$$

This transparency helps identify when predictions rely more on prototypes than demonstrations. The final CoT-enhanced prediction combines reasoning confidence with the base framework's scores:

$$\hat{y}_{CoT}^j = \mathbb{I}\left[\beta \cdot p_{CoT}(y_q^j = 1) + (1 - \beta) \cdot p(y_q^j = 1 | x_q, \mathcal{D}_k) > \tau_j^{thresh}\right] \tag{30}$$

where $\beta$ weights the contribution of reasoning versus direct prediction, typically set to 0.7 to prioritize structured reasoning while maintaining robustness.

The CoT reasoning provides three key advantages for multi-label patent classification: (1) explicit handling of label co-occurrences through conditional reasoning, (2) interpretable decision paths that facilitate error analysis and model improvement, and (3) improved performance on complex patents requiring nuanced technological understanding. By combining the semantic understanding from contrastive embeddings, pattern recognition from retrieved demonstrations, and structured reasoning from GPT-4o, this approach achieves robust multi-label classification with minimal labeled examples while maintaining interpretability. Fig 3 illustrates the whole framework.

## Proposed framework

We now present our complete framework that brings together contrastive learning, retrieval-based demonstration selection, and chain-of-thought reasoning to classify patents across multiple categories. This unified approach requires only a small number of labeled examples while maintaining high accuracy. Algorithm 1 details the step-by-step process from raw patent text to final multi-label predictions.

The framework operates in four integrated phases. First, contrastive pre-training learns domain-specific embeddings that capture multi-label relationships in patent text. Second, retrieval-augmented demonstration selection identifies the most informative examples through multi-faceted similarity scoring. Third, few-shot classification constructs prompts with embedding-guided attention weights. Finally, chain-of-thought reasoning via GPT-4o generates interpretable predictions for each category while handling inter-label dependencies, sparse categories through prototypes, and uncertainty through adaptive thresholding.

This unified approach addresses the key challenges of multi-label patent classification: the annotation bottleneck through few-shot learning, the complexity of technical language through domain-specific pre-training, the multi-label nature through explicit modeling of label co-occurrences, and the need for interpretability through structured reasoning chains. The framework's modular design allows for component improvements while maintaining the overall architecture's effectiveness.

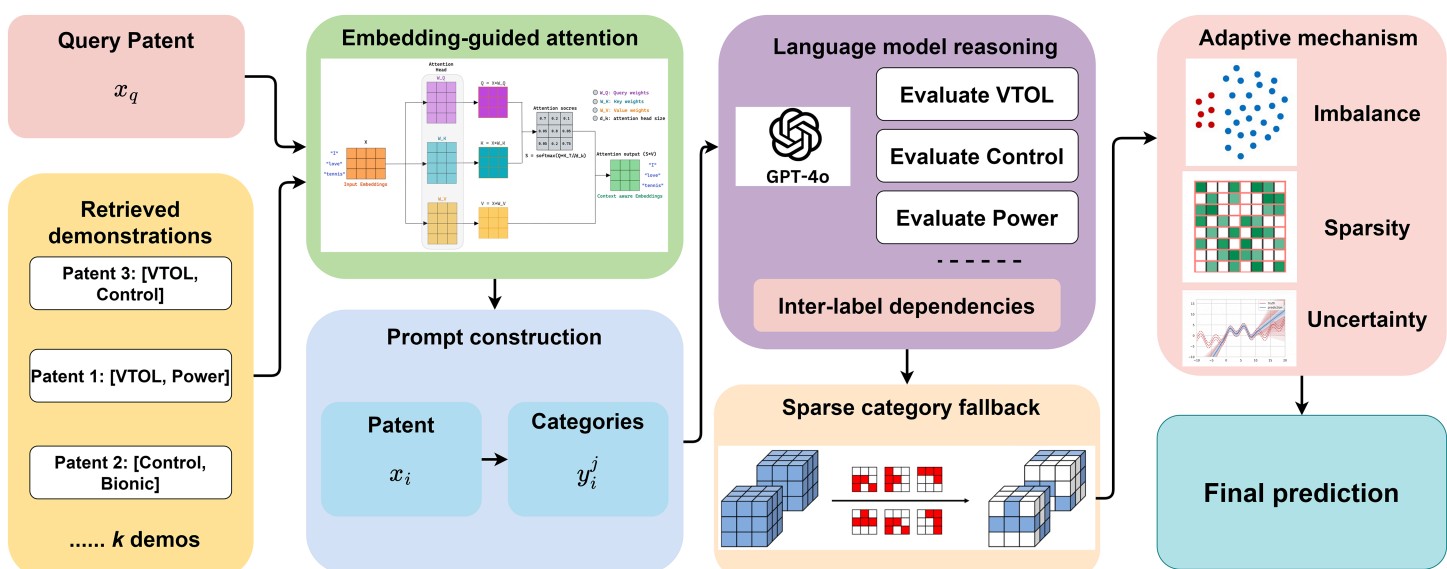

**Fig 3**. **The system combines retrieved demonstrations with embedding-guided attention to perform multi-label patent classification.** The prediction module employs decomposed inference for each category while considering inter-label dependencies, adaptive thresholding based on uncertainty, and prototype-based fallback for sparse categories. The integration of contrastive embeddings, demonstration patterns, and language model reasoning enables robust classification with minimal labeled examples.

**Algorithm 1 Retrieval-augmented contrastive learning for multi-label patent classification.**

1: **Input:** Query patent $x_q$, Patent corpus $\mathcal{D}$, Labeled set $\mathcal{C}$, Categories $\{l_1, ..., l_K\}$
2: **Output:** Multi-label prediction $\hat{y}_q \in \{0, 1\}^K$
3: **// Phase 1: Contrastive Pre-training**
4: Initialize encoder $f_\theta$ with RoBERTa-large weights
5: **for** epoch $= 1$ to $E_{pretrain}$ **do**
6: Sample mini-batch $\mathcal{B} \subset \mathcal{D}$
7: Compute embeddings $\{z_i = f_\theta(x_i)\}_{i \in \mathcal{B}}$
8: Calculate multi-label contrastive loss: $\mathcal{L}_{contrast} = \mathcal{L}_{instance} + \lambda \mathcal{L}_{label}$
9: Update momentum encoder: $\theta' \leftarrow m\theta' + (1 - m)\theta$
10: Update $\theta$ via gradient descent on $\mathcal{L}_{contrast}$
11: **end for**
12: **// Phase 2: Retrieval-Augmented Demonstration Selection**
13: Compute query embedding: $z_q = f_\theta(x_q)$
14: Initialize retrieval set $\mathcal{R} = \varnothing$
15: **for** $j = 1$ to $k$ **do**
16: Compute scores for all $x_i \in \mathcal{C} \setminus \mathcal{R}$:
17: $\text{score}(x_q, x_i) = \alpha_1 \cdot \text{sim}_{sem}(z_q, z_i) + \alpha_2 \cdot \text{sim}_{tech}(x_q, x_i) + \alpha_3 \cdot \text{div}(x_i | \mathcal{R})$
18: Select: $x^* = \arg\max_{x_i} \text{score}(x_q, x_i)$
19: Update: $\mathcal{R} \leftarrow \mathcal{R} \cup \{x^*\}$
20: **end for**
21: Order demonstrations: $\mathcal{D}_k^{ordered} = \text{sort}(\mathcal{R}, \lambda_1 \cdot \text{score} - \lambda_2 \cdot \text{diversity})$
22: **// Phase 3: Few-shot Classification with CoT Reasoning**
23: Construct prompt: $\mathcal{P} = \text{Instruct} \oplus \bigoplus_{i=1}^{k} \text{Demo}(x_i, y_i) \oplus \text{Query}(x_q)$
24: Compute attention weights: $w_i = \text{softmax}(\text{sim}(z_q, z_i)/\tau_{attn} + b_i)$
25: Augment with CoT template: $\mathcal{P}_{CoT} = \mathcal{P} \oplus \text{Reason}_{template}$
26: **// Phase 4: Multi-label Prediction**
27: **for** each category $l_j \in \{l_1, ..., l_K\}$ **do**
28: **// GPT-4o reasoning with API parameters:**
29: temperature=0.3, max_tokens=2048, top_p=0.9, frequency_penalty=0.2
30: Generate reasoning: $\text{CoT}_j = \text{GPT-4o}(\mathcal{P}_{CoT}, l_j, \{y_q^i : i < j\})$
31: Compute CoT probability: $p_{CoT}(y_q^j = 1)$ from reasoning output
32: **// Compute base probability with inter-label dependencies:**
33: $p_{base}(y_q^j = 1) = \sigma(f_{LM}(x_q, \mathcal{D}_k, l_j) + \sum_{m<j} \omega_{jm} \cdot \hat{y}_q^m)$
34: **// Check for sparse category fallback:**
35: **if** $\sum_{i=1}^{k} y_i^j < \epsilon$ **then**
36: Compute prototype: $\text{proto}_j = \frac{1}{|\mathcal{S}_j|} \sum_{x \in \mathcal{S}_j} f_\theta(x)$
37: $p_{fallback}(y_q^j = 1) = \sigma(z_q^T \text{proto}_j / (\|z_q\| \|\text{proto}_j\| \tau_{proto}))$
38: $\alpha_{demo} = \sum_{i=1}^{k} y_i^j / \epsilon$
39: **else**
40: $p_{fallback} = 0, \quad \alpha_{demo} = 1$
41: **end if**
42: **// Adaptive thresholding:**
43: $\tau_j^{thresh} = \tau_0 + \Delta\tau \cdot (\text{freq}(l_j) - \mu_{freq}) + \gamma \cdot \text{uncertainty}(x_q, l_j)$
44: **// Final prediction:**
45: $p_{final} = \beta \cdot p_{CoT} + (1 - \beta) \cdot [\alpha_{demo} \cdot p_{base} + (1 - \alpha_{demo}) \cdot p_{fallback}]$
46: $\hat{y}_q^j = \mathbb{I}[p_{final} > \tau_j^{thresh}]$
47: **end for**
48: **return** $\hat{y}_q = [\hat{y}_q^1, ..., \hat{y}_q^K]$

## Results

### Experimental design

Our experimental evaluation adopts a stratified few-shot protocol that respects the temporal structure of patent data while maintaining representative label distribution across training and testing phases. The complete UAV patent dataset of 15,000 annotated patents undergoes temporal partitioning where patents from 2000-2020 constitute the training corpus (12,000 patents) and patents from 2021-2023 form the test set (3,000 patents). This temporal division reflects realistic deployment scenarios where models trained on historical patents must classify emerging technologies.

Within the training corpus, we establish a few-shot learning protocol following the N-way-K-shot paradigm adapted for multi-label scenarios. For each evaluation episode, we randomly sample $N = 5$ technological categories and select $K \in \{1, 3, 5, 10\}$ demonstration examples per category from the training set, ensuring each selected patent contains at least one target label. The query set consists of 100 patents containing various combinations of the selected categories, with an average of 2.3 labels per patent matching our dataset characteristics.

Categories are grouped into three tiers based on occurrence frequency: frequent categories include VTOL & Hybrid Flight, Surveillance & Mapping, and Flight Control & Stability; moderate categories encompass Modular & Deployable, Endurance & Power Systems, and Structural & Materials; sparse categories comprise Logistics & Cargo, Bionic & Flapping Wing, Specialized Applications, and Multi-Environment. Each evaluation episode samples at least one category from each tier to prevent bias toward frequent categories.

We compare our approach against carefully selected baselines to isolate the contribution of each framework component. Recent patent-specific methods include LLM-AL [29], which combines iterative large language model inference with active learning for scalable multi-label classification, and PatentSBERTa [18], a hybrid approach integrating SBERT sentence embeddings with K-nearest neighbors classification and patent-specific domain adaptation. Traditional multi-label approaches include fine-tuned RoBERTa-Large [5] and XLNet-Large [67], representing current transformer-based methods for patent classification. Few-shot learning baselines comprise Prototypical Networks [68] adapted for multi-label scenarios and META-LSTM [69] with multi-label extensions. Retrieval-augmented approaches include standard RAG using dense passage retrieval with BERT embeddings and RePrompt [70] adapted for multi-label classification.

Our evaluation employs multiple metrics addressing different aspects of multi-label few-shot performance. Macro-F1 score provides equal weighting to all categories:

$$\text{Macro-F1} = \frac{1}{K} \sum_{j=1}^{K} F1_j = \frac{1}{K} \sum_{j=1}^{K} \frac{2 \cdot P_j \cdot R_j}{P_j + R_j} \tag{31}$$

where $P_j$ and $R_j$ represent precision and recall for category $j$, and $K$ denotes the number of categories. Micro-F1 score aggregates predictions across all categories:

$$\text{Micro-F1} = \frac{2 \cdot P_{micro} \cdot R_{micro}}{P_{micro} + R_{micro}} \tag{32}$$

where $P_{micro} = \frac{\sum_{j=1}^{K} TP_j}{\sum_{j=1}^{K} (TP_j + FP_j)}$ and $R_{micro} = \frac{\sum_{j=1}^{K} TP_j}{\sum_{j=1}^{K} (TP_j + FN_j)}$.

Label Ranking Average Precision (LRAP) evaluates the quality of label ranking:

$$\text{LRAP} = \frac{1}{N} \sum_{i=1}^{N} \frac{1}{|Y_i|} \sum_{j \in Y_i} \frac{|\mathcal{L}_{ij}|}{rank_{ij}} \tag{33}$$

where $Y_i$ represents true labels for sample $i$, $\mathcal{L}_{ij} = \{k \in Y_i : score_{ik} \geq score_{ij}\}$, and $rank_{ij}$ denotes the rank of label $j$ for sample $i$. Coverage Error measures the average number of top-ranked labels needed to cover all true labels:

$$\text{Coverage} = \frac{1}{N} \sum_{i=1}^{N} \max_{j \in Y_i} rank_{ij} - 1 \qquad (34)$$

Few-shot adaptation efficiency is measured through learning curves across different shot numbers and category-wise performance analysis for frequent, moderate, and sparse categories. Performance variance across 50 episodes per configuration provides robustness assessment. Demonstration retrieval effectiveness uses Precision@K and Recall@K metrics that measure whether retrieved patents contain relevant technological categories. Chain-of-thought reasoning quality undergoes human expert assessment where three domain experts rate 500 randomly selected reasoning chains on 5-point Likert scales across coherence, factual accuracy, and decision support utility dimensions. Statistical significance is assessed through paired t-tests with Bonferroni correction for multiple comparisons. All experiments are conducted on a single NVIDIA GeForce RTX 4090 GPU with 32GB memory.

## Overall performance comparison

Our proposed framework achieves meaningful improvements across all evaluation metrics compared to existing approaches. Table 3 presents the comprehensive performance comparison under the 5-shot setting, which represents a challenging few-shot scenario with limited demonstration examples per category.

The results demonstrate consistent and statistically significant improvements across all metrics. Our framework achieves a Macro-F1 score of 0.847 (±0.021), which represents improvements of 16.2% over RoBERTa-Large and 14.3% over XLNet-Large. Compared to recent patent-specific methods, our framework achieves 6.1% improvement over LLM-AL and 11.2% improvement over PatentSBERTa in Macro-F1 performance. The Micro-F1 performance reaches 0.892 (±0.018), with improvements of 11.4% and 9.4% over RoBERTa and XLNet respectively. These improvements are particularly notable given the challenging nature of multi-label patent classification and the limited annotation requirements of the few-shot setting.

Label ranking performance, measured by LRAP, shows our framework achieves 0.878 (±0.019). This outperforms transformer baselines by margins that exceed 14%. Our framework also outperforms LLM-AL by 5.3% and PatentSBERTa by 12.3% on LRAP. Coverage Error, where lower values indicate better performance, demonstrates our

**Table 3**. **Overall performance comparison under 5-shot setting.**

| Method | Macro-F1 | Micro-F1 | LRAP | Coverage |
|---|---|---|---|---|
| Our Framework | **0.847±0.021** | **0.892±0.018** | **0.878±0.019** | **1.23±0.087** |
| *Recent Patent-Specific Methods* | | | | |
| LLM-AL [29] | 0.798±0.028 | 0.865±0.024 | 0.834±0.025 | 1.41±0.112 |
| PatentSBERTa [18] | 0.762±0.033 | 0.828±0.030 | 0.782±0.031 | 1.68±0.135 |
| *General Few-Shot & Transformer Baselines* | | | | |
| XLNet-Large [67] | 0.741±0.031 | 0.815±0.027 | 0.768±0.028 | 1.74±0.128 |
| RoBERTa-Large [66] | 0.729±0.034 | 0.801±0.029 | 0.756±0.032 | 1.87±0.142 |
| RePrompt [70] | 0.716±0.035 | 0.773±0.031 | 0.742±0.033 | 1.92±0.148 |
| RAG+BERT [44] | 0.698±0.038 | 0.758±0.033 | 0.715±0.035 | 2.08±0.154 |
| Prototypical [68] | 0.652±0.042 | 0.724±0.037 | 0.687±0.039 | 2.31±0.167 |
| META-LSTM [69] | 0.634±0.046 | 0.712±0.041 | 0.671±0.043 | 2.45±0.183 |

Results represent mean ± standard deviation across 50 episodes. Bold indicates best performance. Statistical significance was evaluated through paired t-tests with Bonferroni correction ($\alpha = 0.00625$ for eight comparisons). All improvements by our framework are statistically significant ($p < 0.001$). Effect sizes (Cohen's $d$) range from 1.2 to 2.3, which demonstrates both statistical and practical significance.

framework's superior label ranking capabilities with a score of 1.23 (±0.087), compared to 1.87 for RoBERTa-Large and 1.41 for LLM-AL.

Statistical significance was evaluated through paired t-tests that compared our framework against each baseline method across all four metrics. We conducted pairwise comparisons for all eight baseline methods. This yielded 32 total statistical tests (8 baselines × 4 metrics). To control for family-wise error rate in multiple comparisons, we applied Bonferroni correction with significance threshold $\alpha = 0.05/8 = 0.00625$. All 32 comparisons yielded p-values below 0.001, which substantially exceeded the corrected significance threshold. The observed improvements are therefore highly unlikely to occur by chance. Effect sizes, measured by Cohen's $d$, range from 1.2 to 2.3 across comparisons. These values indicate large practical significance. The combined statistical and effect size analyses confirm that our framework provides meaningful and robust improvements over existing approaches, including recent patent-specific methods.

Fig 4 illustrates the comprehensive performance comparison with error bars representing standard deviations across 50 experimental episodes. The visualization clearly demonstrates our framework's consistent superiority across diverse evaluation dimensions.

### Few-shot learning effectiveness

The few-shot learning curves in Fig 5 reveal the framework's superior data efficiency across varying numbers of demonstration examples. Performance consistently improves as the number of shots increases from 1 to 10, with our framework maintaining substantial advantages at all shot levels.

In the challenging 1-shot scenario, our framework achieves Macro-F1 and Micro-F1 scores of 0.723 and 0.789 respectively, representing improvements of 23.2% and 21.2% over RoBERTa-Large, 18.5% and 15.7% over LLM-AL, and 25.8% and 22.3% over PatentSBERTa. While absolute performance increases with additional shots, the relative improvements narrow as baseline methods also benefit from more demonstrations, reaching 15.3% and 10.0% improvements over RoBERTa-Large in the 10-shot setting. The learning curves demonstrate effective knowledge transfer from contrastive pre-training, enabling rapid adaptation with minimal supervision.

The error bars reveal that our framework maintains more stable performance across episodes, with standard deviations consistently lower than baseline methods. This stability indicates robust generalization capabilities, crucial for practical deployment scenarios where training data is limited and variable.

### Computational efficiency

Table 4 compares computational efficiency between our framework and baseline methods across training and inference phases. Our framework adds 2M parameters (a 0.6% increase) over RoBERTa-Large through projection layers and retrieval components. This represents minimal model size overhead. The contrastive pre-training phase demands substantially fewer computational resources than full supervised fine-tuning employed by transformer baselines (8.2 hours versus 24-26.5 hours) and achieves training efficiency comparable to patent-specific methods (8.2 hours versus 18-22 hours for LLM-AL and PatentSBERTa) on a single RTX 4090 GPU. This advantage arises because our framework leverages unlabeled patent corpora during pre-training while requiring only minimal labeled demonstrations during inference.

Inference latency consists of two components: local model processing (48-52ms per patent) and GPT-4o API calls (165-195ms per patent with typical network latency). The GPT-4o integration introduces additional inference time. However, this overhead proves justified given the substantial performance improvements and interpretability benefits. For deployment scenarios that demand lower latency, the chain-of-thought reasoning module can be disabled with graceful performance degradation (5.4% Macro-F1 reduction as shown in ablation analysis). This modification

**Fig 4. Overall performance comparison across evaluation metrics.** The proposed framework consistently outperforms baseline methods across Macro-F1, Micro-F1, LRAP, and Coverage Error metrics. Error bars represent standard deviations across 50 experimental episodes. Statistical significance indicators (***$p < 0.001$, **$p < 0.01$, *$p < 0.05$) show comparisons against our framework.

reduces total inference time to baseline levels while preserving advantages from contrastive pre-training and semantic retrieval.

The memory footprint during inference remains comparable to baseline methods at 3.3GB GPU memory, since GPT-4o operates through API calls without local parameter storage. This efficient resource utilization enables deployment on standard research-grade GPUs. The framework achieves superior accuracy-efficiency trade-offs: 16.2% higher Macro-F1 than RoBERTa-Large with 66% training time reduction, 6.1% improvement over LLM-AL with 54% less training time, and 11.2% improvement over PatentSBERTa with 63% training time reduction. Despite slightly longer inference times due to GPT-4o integration (228ms versus 245ms for LLM-AL and 65ms for PatentSBERTa), the substantial accuracy gains justify this overhead. These characteristics render our approach particularly suitable for resource-constrained settings where both annotation budgets and computational resources remain limited.

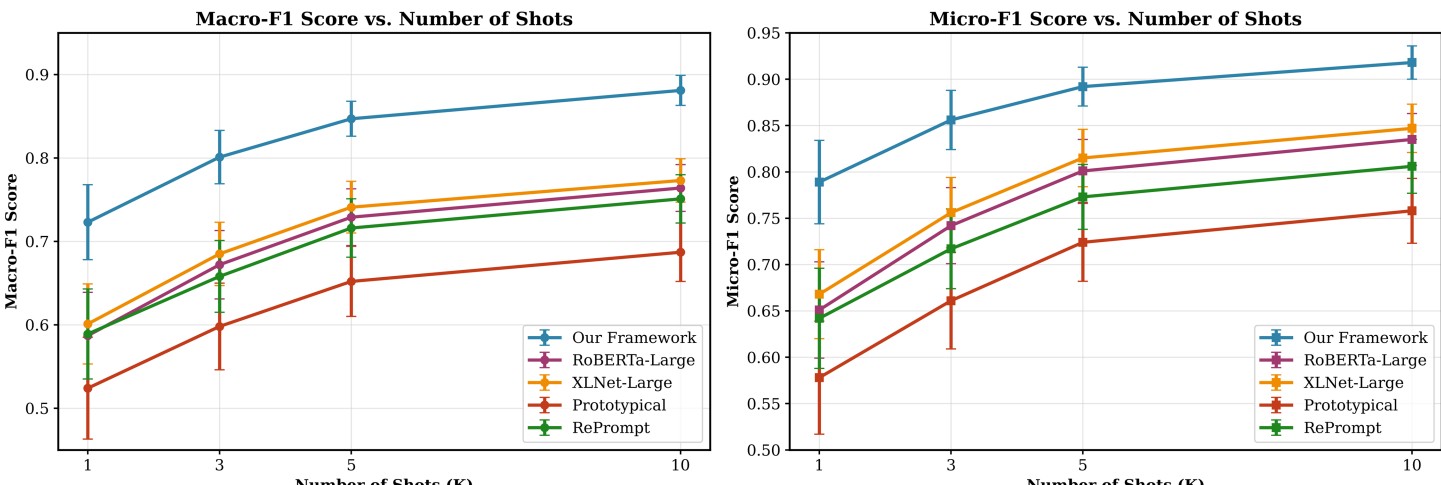

**Fig 5. Few-shot learning curves showing performance vs. number of shots.** Our framework demonstrates superior few-shot learning capabilities across all shot settings. Left: Macro-F1 performance prioritizes rare category detection. Right: Micro-F1 performance emphasizes overall classification accuracy. Error bars represent standard deviations across 50 episodes.

**Table 4. Computational efficiency comparison.**

| Method | Params (M) | GFLOPs | Inference (ms) | Memory (GB) | Training (h) |
|---|---|---|---|---|---|
| Our Framework | 357 | 12.4 | 48 + 180[†] | 3.3 | 8.2[‡] |
| *Patent-Specific Methods* | | | | | |
| LLM-AL | 110 | 5.2 | 45 + 200[§] | 2.8 | 18.0[¶] |
| PatentSBERTa | 355 | 12.3 | 50 + 15[#] | 3.4 | 22.0 |
| *General Baselines* | | | | | |
| RoBERTa-Large | 355 | 12.1 | 45 | 3.2 | 24.0 |
| XLNet-Large | 340 | 13.2 | 48 | 3.1 | 26.5 |
| RePrompt | 355 | 12.1 | 48 + 35[*] | 3.3 | 14.0 |
| RAG+BERT | 110 | 4.8 | 38 + 25[*] | 1.5 | 12.0 |
| Prototypical | 355 | 12.1 | 52 | 3.4 | 6.5[**] |
| META-LSTM | 110 | 3.8 | 42 | 1.8 | 5.5[**] |

[†]Local processing (48ms) + GPT-4o API calls (180ms avg). CoT module can be disabled for lower latency. [‡]Contrastive pre-training time; transformer baselines require full supervised fine-tuning. [§]Encoder inference (45ms) + iterative LLM API calls with active learning (200ms avg). [¶]Iterative active learning with multiple query rounds. [#]SBERT embedding (50ms) + KNN retrieval (15ms). [*]Retrieval overhead for demonstration selection. [**]Meta-learning across episodes; no pre-training phase. All measurements on single NVIDIA RTX 4090 GPU with batch size 32. GFLOPs computed for single patent inference. Memory indicates peak GPU usage during training.

## Ablation study

The ablation study in Table 5 evaluates each framework component's contribution to overall performance. Removing individual components reveals their relative importance and validates the integrated design approach.

Contrastive pre-training contributes most significantly to performance, with its removal causing a 6.8% drop in Macro-F1 (from 0.847 to 0.789). This substantial decrease demonstrates the importance of domain-adapted representations that capture multi-label co-occurrence patterns in patent text. The semantic retrieval mechanism provides the second-largest contribution, with random demonstration selection causing a 9.7% performance drop (Macro-F1: 0.765). This validates our multi-faceted similarity scoring approach for identifying informative demonstrations.

**Table 5**. Ablation study: component contribution analysis.

| Configuration | Macro-F1 | Micro-F1 | LRAP | Coverage | Contribution |
|---|---|---|---|---|---|
| Full Framework | **0.847±0.021** | **0.892±0.018** | **0.878±0.019** | **1.23±0.087** | – |
| w/o Contrastive Pre-training | 0.789±0.028 | 0.834±0.024 | 0.821±0.028 | 1.65±0.142 | −6.8% |
| w/o Semantic Retrieval | 0.765±0.032 | 0.812±0.028 | 0.798±0.032 | 1.89±0.167 | −9.7% |
| w/o Chain-of-Thought | 0.801±0.025 | 0.856±0.021 | 0.845±0.025 | 1.47±0.128 | −5.4% |
| w/o Inter-label Dependency | 0.823±0.023 | 0.871±0.019 | 0.863±0.023 | 1.38±0.098 | −2.8% |

Results represent mean ± standard deviation across 20 episodes. Bold indicates full framework performance. Contribution shows Macro-F1 performance drop percentage when each component is removed. w/o Contrastive: standard RoBERTa embeddings without domain-adaptive contrastive pre-training. w/o Retrieval: random demonstration selection instead of multi-faceted similarity scoring. w/o CoT: direct prediction without chain-of-thought reasoning. w/o Inter-label: binary relevance approach treating labels independently.

Chain-of-thought reasoning contributes a 5.4% improvement (Macro-F1 drop from 0.847 to 0.801 without CoT), which demonstrates the value of structured reasoning for complex multi-label decisions. The inter-label dependency modeling provides a 2.8% improvement. This shows benefits from explicitly modeling technological co-occurrence patterns rather than treating labels independently.

Coverage error improvements follow similar patterns, with contrastive pre-training and semantic retrieval showing the largest contributions. The cumulative effect of all components results in the full framework's superior performance, which validates the integrated design approach.

## Discussion

Our experimental findings reveal that the integration of contrastive learning with retrieval-augmented few-shot learning offers an effective pathway for multi-label patent classification under annotation constraints. Rather than relying on extensive expert annotation, our approach demonstrates meaningful improvements across evaluation metrics, particularly those addressing the complexities of scaling patent classification to emerging technological domains.

We observe that contrastive pre-training effectiveness in the patent domain stems from its capacity to capture intricate co-occurrence patterns within technological innovations. Indeed, patents describing integrated systems naturally span multiple categories, and our multi-label contrastive objective learns representations that respect these technological relationships. This domain adaptation proves especially valuable for underrepresented categories, whereas traditional fine-tuning approaches encounter difficulties due to limited examples.

Our findings suggest that retrieval-augmented demonstration selection plays a pivotal role in few-shot performance. The multi-faceted similarity scoring mechanism, which combines semantic embeddings with technical domain features, enables identification of relevant demonstrations that guide classification decisions. Instead of naive similarity-based approaches, the diversity constraints prevent redundant retrievals while ensuring comprehensive label space coverage.

The chain-of-thought reasoning integration offers both performance enhancements and interpretability benefits. The structured reasoning paths provide insights into classification decisions, facilitating error analysis and model debugging. However, we note that the computational overhead of GPT-4o inference presents practical considerations for large-scale deployment scenarios.

We acknowledge several limitations that warrant consideration. Our evaluation focuses specifically on UAV patent classification, and generalization to other technological domains or patent categories requires additional empirical validation. While the UAV domain provides an excellent testbed due to its multidisciplinary nature and representative multi-label complexity, demonstrating broader applicability across diverse technical fields remains a direction for future work. The framework's reliance on GPT-4o for chain-of-thought reasoning introduces computational costs and API dependencies that may limit deployment in resource-constrained settings. Furthermore, the contrastive pre-training phase requires domain-specific patent corpora, which may necessitate additional data collection efforts when extending to new technical

domains. The reliance on English translations may introduce noise for patents originally filed in other languages. Additionally, although the temporal evaluation split reflects realistic scenarios, it may not fully capture challenges in classifying patents describing genuinely novel technological concepts absent from historical data.

The framework's modular architecture provides promising directions for future enhancement. Alternative contrastive objectives, particularly those incorporating hierarchical contrastive learning that explicitly models patent classification taxonomies, could improve representation quality. Furthermore, the retrieval mechanism could benefit from learned similarity metrics optimized specifically for patent demonstration selection. In principle, integration with domain-specific language models beyond GPT-4o may provide performance gains while reducing computational costs. Future research will address these limitations by evaluating the framework across multiple patent categories and other specialized document classification domains, exploring lightweight alternatives to large language models, and investigating transfer learning strategies that reduce domain-specific data requirements.

## Conclusion

We address the challenge of multi-label patent classification under minimal annotation constraints through an integrated approach combining contrastive learning, retrieval-augmented demonstration selection, and chain-of-thought reasoning. Our framework demonstrates meaningful improvements over conventional approaches, achieving Macro-F1 and Micro-F1 scores of 0.847 and 0.892 respectively on UAV patent classification tasks.

Our contributions encompass a multi-label contrastive pre-training strategy that captures technological co-occurrence patterns, a retrieval mechanism that identifies informative demonstrations through multi-faceted similarity scoring, and a structured reasoning approach that handles inter-label dependencies while maintaining interpretability. The systematic ablation analysis validates each component's importance, with contrastive pre-training and semantic retrieval providing the most pronounced performance contributions.

We demonstrate the framework's effectiveness across varying numbers of demonstration examples, from challenging 1-shot to more supportive 10-shot scenarios, which highlights its practical utility for real-world patent classification tasks. The consistent performance advantages over established baselines, including recent retrieval-augmented approaches, suggest that our integrated design addresses the unique challenges of multi-label few-shot learning in technical domains.

Beyond patent classification, our approach offers broader implications for few-shot learning in specialized domains where obtaining expert annotations remains prohibitively expensive. The combination of domain-adapted representations, intelligent demonstration selection, and structured reasoning provides a general framework applicable to other technical document classification tasks. In particular, those requiring minimal supervision while maintaining accuracy and interpretability may benefit from this integrated methodology.

## Author contributions

**Data curation:** Wenlong Zheng, Xin Li.

**Formal analysis:** Wenlong Zheng.

**Funding acquisition:** Shikun Chen.

**Investigation:** Guoqing Cui, Shikun Chen.

**Methodology:** Xin Li, Guoqing Cui, Shikun Chen.

**Project administration:** Xin Li.

**Resources:** Wenlong Zheng, Guoqing Cui.

**Software:** Wenlong Zheng.

**Supervision:** Wenlong Zheng.

**Validation:** Xin Li, Shikun Chen.

**Visualization:** Wenlong Zheng, Shikun Chen.

**Writing – original draft:** Wenlong Zheng, Shikun Chen.

**Writing – review & editing:** Wenlong Zheng, Guoqing Cui, Shikun Chen.

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
