## [Decision Letter · Decision Letter 0]

7 Nov 2025

PONE-D-25-47197

Contrastive learning enhanced retrieval-augmented few-shot framework for multi-label patent classification

PLOS ONE

Dear Dr.  Chen,

Thank you for submitting your manuscript to PLOS ONE. After careful consideration, we feel that it has merit but does not fully meet PLOS ONE’s publication criteria as it currently stands. Therefore, we invite you to submit a revised version of the manuscript that addresses the points raised during the review process.

We look forward to receiving your revised manuscript.

Kind regards,

Haofeng Zhang

Academic Editor

PLOS ONE

Journal Requirements:

2. Please amend the manuscript submission data (via Edit Submission) to include author “Wenlong Zheng”.

3. Please amend your authorship list in your manuscript file to include author “Shikun Chen”.

Reviewer's Responses to Questions

**Comments to the Author**

1. Is the manuscript technically sound, and do the data support the conclusions?

Reviewer #1: Yes

Reviewer #2: Yes

2. Has the statistical analysis been performed appropriately and rigorously?

Reviewer #1: Yes

Reviewer #2: No

3. Have the authors made all data underlying the findings in their manuscript fully available?

Reviewer #1: Yes

Reviewer #2: Yes

4. Is the manuscript presented in an intelligible fashion and written in standard English?

Reviewer #1: Yes

Reviewer #2: Yes

5. Review Comments to the Author

Reviewer #1: Summary:

This paper proposes a retrieval-augmented few-shot framework for multi-label patent classification that combines domain-specific contrastive pre-training with retrieval-enhanced few-shot learning for label decisions. Concretely, the authors build multi-label–aware contrastive objectives, rank demonstrations with a composite similarity, and then perform decomposed, category-wise predictions with adaptive thresholds. Evaluated on a curated dataset, the method reports improvements of Macro-F1/Micro-F1 over few-shot and transformer baselines.

Weakness:

- The evaluation is confined to a single domain with a custom 10-category schema. Results may not transfer to other technical domains. Evaluating on other domains would strengthen claims of generality.

- Efficiency is under-reported. The paper should provide efficiency analysis like a parameter/GFLOPs comparison to baselines.

- The figure is very vague. The authors need to refine the resolution of the figures to make it readable.

- Missing related work. Several recent works relevant to few-shot learning are relevant and should be cited:

a. Multimodality Helps Few-shot 3D Point Cloud Semantic Segmentation (ICLR 2025)

b. Generalized Few-shot 3D Point Cloud Segmentation with Vision-Language Model (CVPR 2025)

c. In-Context Learning for Text Classification with Many Labels (ACL 2023)

Reviewer #2: 1.The "Background and related work" section covers too many research directions (six in total) and employs an overly complex classification scheme. It is recommended to streamline and consolidate this part for better clarity and focus.

2.The "Dataset" subsection should clarify whether the dataset is a novel contribution of this work or was obtained from an existing source, providing appropriate citations in either case.

3.Figures should be embedded close to their corresponding references in the main text rather than being placed later in the document. Additionally, the current image resolution is insufficient and should be improved.

4.For Table 3, please include citations for the comparative algorithms listed. Furthermore, the note stating "All improvements by our framework are statistically significant (p < 0.001)" should be supported with relevant details in the text explaining where and how statistical significance was evaluated, as well as what this implies for the results.

5.It is recommended to present the ablation study results in a tabular format, which would more clearly demonstrate the contribution of each individual component.

6.Among the six algorithms currently included in “Overall performance comparison” section, it is necessary to supplement with 2–3 recently proposed algorithms specifically designed for patent classification.

7.The references contain formatting inconsistencies. Journal and conference names should follow standard capitalization rules (e.g., each major word capitalized). Additionally, page numbers are missing in several entries and must be provided.

6. PLOS authors have the option to publish the peer review history of their article (what does this mean?). If published, this will include your full peer review and any attached files.

Reviewer #1: No

Reviewer #2: No

---

## [Author Response · Author response to Decision Letter 1]

11 Dec 2025

We would like to sincerely thank the reviewers for their helpful remarks and corrections. Our response to the raised points can be found below. We hope that the revised manuscript addresses all issues in a satisfying manner; all changes made to the article are marked with blue highlighting.

Reviewer #1 comments

1.“ The evaluation is confined to a single domain with a custom 10-category schema. Results may not transfer to other technical domains. Evaluating on other domains would strengthen claims of generality. ”

We sincerely thank the reviewer for this important observation. We acknowledge that our evaluation is confined to UAV patent classification, and we agree that broader evaluation would strengthen generalizability claims. We selected the UAV domain deliberately as it exemplifies multi-label classification challenges through its inherent multidisciplinary nature, encompassing mechanical, electronic, software, and commu- nication technologies within individual patents. This characteristic makes it an ideal testbed for validating our approach under realistic multi-label conditions.

In response to this valuable feedback, we have added a comprehensive limitations paragraph in the Discussion section that explicitly acknowledges the domain-specific evaluation and discusses the need for additional validation across other technological domains. The limitations discussion addresses (1) the need for cross-domain validation beyond UAV patents, (2) computational costs and API dependencies from GPT-4o in- tegration, (3) domain-specific data requirements for contrastive pre-training, and (4) challenges with non-English patents and genuinely novel technologies. We also out- line future research directions including evaluating across multiple patent categories, exploring lightweight LLM alternatives, and investigating transfer learning strategies to reduce domain-specific data requirements.

While our current focus remains on demonstrating the framework’s effectiveness within the patent domain, we believe the modular architecture and methodological contribu- tions provide a foundation for future cross-domain evaluation.

2.“ Efficiency is under-reported. The paper should provide efficiency analysis like a parame- ter/GFLOPs comparison to baselines. ”

We thank the reviewer for highlighting this important aspect. We have added a ded- icated ”Computational efficiency” subsection in the Results section with comprehen- sive efficiency analysis. Table 4 presents detailed comparisons including parameters, GFLOPs, inference time, GPU memory, and training time across all eight baseline methods, organized into patent-specific methods (LLM-AL, PatentSBERTa) and gen- eral baselines (RoBERTa-Large, XLNet-Large, RePrompt, RAG+BERT, Prototypi- cal, META-LSTM).

3.“The figure is very vague. The authors need to refine the resolution of the figures to make it readable.”

We appreciate this feedback. Following PLOS ONE submission guidelines, figures in the LaTeX manuscript are placeholders, as the template requires figures to be uploaded separately. We have provided all figures as high-resolution (600 DPI) files in the revised submission package to ensure optimal readability.

4.“

Missing related work. Several recent works relevant to few-shot learning are relevant and should be cited:

(a) Multimodality Helps Few-shot 3D Point Cloud Semantic Segmentation (ICLR 2025)

(b) Generalized Few-shot 3D Point Cloud Segmentation with Vision-Language Model (CVPR 2025)

(c) In-Context Learning for Text Classification with Many Labels (ACL 2023) ”

We thank the reviewer for these valuable suggestions. We have incorporated all three recommended citations into the ”Retrieval-augmented few-shot learning” subsection. Specifically, we cite Milios et al. (2023) on in-context learning for multi-label text classification, which directly relates to our demonstration selection approach. We also cite An et al. (2024, 2025) on few-shot 3D point cloud segmentation to acknowledge recent advances in few-shot learning across computer vision domains, noting that cross- domain insights may inform future methodological developments.

Reviewer #2 comments

1.“ The ”Background and related work” section covers too many research directions (six in total) and employs an overly complex classification scheme. It is recommended to streamline and consolidate this part for better clarity and focus. ”

We appreciate this constructive feedback. We have streamlined the Background and re- lated work section from six subsections to four by consolidating related content. Specif- ically, we merged ”Multi-label patent classification,” ”Domain-specific pre-training and technical language models,” and ”UAV patent classification” into a single uni- fied subsection titled ”Patent classification with domain-specific language models.” This consolidation integrates patent-specific challenges, domain adaptation strategies, and UAV testbed justification into a coherent narrative. The revised structure main- tains focus on the four core methodological pillars: patent-specific models, contrastive learning, retrieval-augmented few-shot learning, and chain-of-thought reasoning.

2.“ The ”Dataset” subsection should clarify whether the dataset is a novel contribution of this work or was obtained from an existing source, providing appropriate citations in either case. ”

We thank the reviewer for this important clarification request. We have explicitly stated in the Dataset subsection that the curated UAV patent dataset with 15,000 expert-annotated patents across ten technological categories represents a novel contri- bution of this work. We have also added a footnote with the GitHub repository URL for dataset access and citation.

3.“ Figures should be embedded close to their corresponding references in the main text rather than being placed later in the document. Additionally, the current image resolution is in- sufficient and should be improved. ”

We appreciate this suggestion. Following PLOS ONE submission guidelines, we have repositioned all figures to appear immediately after their first citation in the text. All figures have been provided as high-resolution (600 DPI) files in the revised submission package.

4.“For Table 3, please include citations for the comparative algorithms listed. Furthermore, the note stating ”All improvements by our framework are statistically significant (p < 0.001)” should be supported with relevant details in the text explaining where and how sta- tistical significance was evaluated, as well as what this implies for the results.”

We thank the reviewer for this important observation. We have added citations for all baseline algorithms in Table 3 and substantially expanded the statistical significance explanation in both the revised manuscript and table footnote to include detailed methodology, multiple comparison correction, and effect sizes.

5.“It is recommended to present the ablation study results in a tabular format, which would more clearly demonstrate the contribution of each individual component.”

We appreciate this suggestion. We have presented the ablation study results in a tabular format (now Table 5) in the ”Ablation study” subsection.

6.“Among the six algorithms currently included in ”Overall performance comparison” section, it is necessary to supplement with 2–3 recently proposed algorithms specifically designed for patent classification.”

We appreciate this valuable suggestion. We have supplemented the comparison with two recently proposed patent-specific methods: (1) LLM-AL (Xiong et al., 2025), an iterative large language model with active learning, and (2) PatentSBERTa (Bekamiri et al., 2024), a hybrid SBERT-KNN model with patent-specific domain adaptation. We reorganized Table 3 to distinguish recent patent-specific methods from general baselines, updated statistical tests for eight comparisons, and added analysis demon- strating our framework’s improvements over these state-of-the-art patent classification approaches.

7.“The references contain formatting inconsistencies. Journal and conference names should follow standard capitalization rules (e.g., each major word capitalized). Additionally, page numbers are missing in several entries and must be provided.”

We thank the reviewer for this careful observation. We have corrected all formatting inconsistencies in the bibliography: (1) Capitalized journal names following standard rules (”arXiv Preprint,” ”IEEE Access,” ”Advances in Neural Information Processing Systems”), (2) For entries lacking page numbers, we have either added them where available or appropriately classified entries as working papers or theses where page numbers do not apply. All journal articles now include complete bibliographic infor- mation.

Again, we are grateful to the reviewers for their time and effort, and we sincerely thank them for their valuable help in improving our manuscript.

Best regards, on behalf of the authors,

Shikun Chen

---

## [Decision Letter · Decision Letter 1]

4 Jan 2026

Contrastive learning enhanced retrieval-augmented few-shot framework for multi-label patent classification

PONE-D-25-47197R1

Dear Dr. Chen,

We’re pleased to inform you that your manuscript has been judged scientifically suitable for publication and will be formally accepted for publication once it meets all outstanding technical requirements.

Kind regards,

Haofeng Zhang

Academic Editor

PLOS One

Additional Editor Comments (optional):

Reviewers' comments:

Reviewer's Responses to Questions

**Comments to the Author**

1. If the authors have adequately addressed your comments raised in a previous round of review and you feel that this manuscript is now acceptable for publication, you may indicate that here to bypass the “Comments to the Author” section, enter your conflict of interest statement in the “Confidential to Editor” section, and submit your "Accept" recommendation.

Reviewer #1: All comments have been addressed

Reviewer #2: All comments have been addressed

2. Is the manuscript technically sound, and do the data support the conclusions?

Reviewer #1: Yes

Reviewer #2: Yes

3. Has the statistical analysis been performed appropriately and rigorously?

Reviewer #1: Yes

Reviewer #2: Yes

4. Have the authors made all data underlying the findings in their manuscript fully available?

Reviewer #1: Yes

Reviewer #2: Yes

5. Is the manuscript presented in an intelligible fashion and written in standard English?

Reviewer #1: Yes

Reviewer #2: Yes

6. Review Comments to the Author

Reviewer #1: (No Response)

Reviewer #2: This revised work is significantly improved and demonstrates substantial refinement. l recommend acceptance of the work in the current form.

7. PLOS authors have the option to publish the peer review history of their article (what does this mean?). If published, this will include your full peer review and any attached files.

Reviewer #1: No

Reviewer #2: No

---

## [Editor Report · Acceptance letter]

PONE-D-25-47197R1

PLOS One

Dear Dr. Chen,

I'm pleased to inform you that your manuscript has been deemed suitable for publication in PLOS One. Congratulations! Your manuscript is now being handed over to our production team.

Kind regards,

on behalf of

Professor Haofeng Zhang

Academic Editor

PLOS One